# A Real-World Cost-Effectiveness Analysis of Rivaroxaban versus Vitamin K Antagonists for the Treatment of Symptomatic Venous Thromboembolism: Lessons from the REMOTEV Registry

**DOI:** 10.3390/medicina59010181

**Published:** 2023-01-16

**Authors:** Sabrina Kepka, Elena-Mihaela Cordeanu, Kevin Zarca, Anne-Sophie Frantz, Patrick Ohlmann, Emmanuel Andres, Pascal Bilbault, Isabelle Durand-Zaleski, Dominique Stephan

**Affiliations:** 1Emergency Department, Strasbourg Regional University Hospital, 1 Place de l’Hôpital, 67091 Strasbourg, France; 2ICube, UMR 7357, CNRS, 300 Bd Sébastien Brant, 67400 Illkirch-Graffenstaden, France; 3Department of Hypertension, Vascular Disease and Clinical Pharmacology, Strasbourg Regional University Hospital, 67091 Strasbourg, France; 4UMR 1260, INSERM Regenerative Nanomedecine, CRBS, 1 Rue Eugène Boeckel, 67000 Strasbourg, France; 5URCEco, Hôtel Dieu, AP-HP, 1 Place du Parvis Notre Dame, 75004 Paris, France; 6Cardiology Department, Strasbourg Regional University Hospital, 67091 Strasbourg, France; 7Internal Medicine Department, Strasbourg Regional University Hospital, 67091 Strasbourg, France; 8Translational Medicine Federation, University of Strasbourg, 67084 Strasbourg, France; 9CRESS, INSERM, INRA , Université Paris-Est Créteil (UPEC), 94000 Créteil, France

**Keywords:** venous thromboembolism, pulmonary embolism, cost-effectiveness, direct oral anticoagulant, vitamin K antagonist, rivaroxaban

## Abstract

*Background and objectives*: Venous thromboembolism (VTE) represents a health and economic burden with consequent healthcare resource utilization. Direct oral anticoagulants (DOACs) have emerged as the mainstay option for VTE treatment but few data exist on their cost-effectiveness as compared to the standard therapy (vitamin K antagonists (VKAs)). This study aimed to assess the cost-effectiveness of rivaroxaban compared to VKAs in VTE treatment by calculating the incremental cost effectiveness ratio (ICER). *Materials and methods*: We conducted a prospective observational study based on the REMOTEV registry, including patients hospitalized for VTE from 23 October 2013 to 31 July 2015, to evaluate the impact of the anticoagulant treatment (DOACs versus VKAs) on 6-month complications: major or clinically relevant non-major bleeding, VTE recurrence and all-cause death. Rivaroxaban was the only DOAC prescribed in this study. The ICER was calculated as the difference in costs divided by the difference in effectiveness. *Results*: Among the 373 patients included, 279 were treated with rivaroxaban (63.1 ± 17.9 years old; 49% men) and 94 with VKAs (71.3 ± 16.6 years old; 46% men). The mean cost was EUR 5662 [95% CI 6606; 9060] for rivaroxaban and EUR 7721 [95% CI 5130; 6304] for VKAs, while effectiveness was 0.0586 95% CI [0.0114; 0.126] for DOACs and 0.0638 [95% CI 0.0208; 0.109] for VKAs. The rivaroxaban treatment strategy was dominant with costs per patient EUR 2059 lower [95% CI −3582; −817] and a higher effectiveness of 0.00527 [95% CI −0.0606; 0.0761] compared to VKAs. *Conclusions*: This study provides real-world evidence that rivaroxaban is not only an efficient and safe alternative to VKAs for eligible VTE patients, but also cost-saving.

## 1. Introduction

Venous thromboembolism (VTE), including deep vein thrombosis (DVT) and pulmonary embolism (PE), is the third most frequent acute cardiovascular syndrome with an incidence of 104 to 183 per 100,000 person-years in Europe [1]. Certain risk factors have been associated with higher risk of VTE occurrence: immobilization, surgery, active cancer, estrogen-based therapies, certain psychiatric conditions, acute and chronic inflammatory states and, more recently, COVID-19 infection [2,3,4]. Oral anticoagulants (direct oral anticoagulants (DOACs) or vitamin K antagonists (VKAs)) are the mainstay of VTE treatment [5,6]. Before DOACs’ advent, VKAs were the gold standard for oral anticoagulation, but they have since become a second-line, alternative treatment as their use was associated with a higher bleeding risk (2–5% per year of major bleeding), related to their narrow therapeutic range, requiring regular laboratory monitoring [7]. The current international guidelines recommend DOACs as the first-choice treatment for VTE, with a few exceptions: severe renal impairment, pregnancy and lactation, gastrointestinal and urothelial cancer, and antiphospholipid syndrome [2,8,9,10,11,12,13,14].

VTE is responsible for a high premature morbidity and mortality related to either thrombosis recurrence or anticoagulant treatment complications with 30-day and 1-year historical survival rates of 72% and 63%, respectively [15]. Thus, VTE represents an economic burden with consequent healthcare resource utilization and high costs due to hospital readmission and treatment with a total healthcare expenditure estimated at EUR 8.5 billion in the European Union [16]. Drug costs for VKAs are low but total costs are notably impacted by the requirement for therapeutic monitoring and the costs of emergent hospital management in case of major bleeding [17]. Although DOACs’ acquisition cost per patient is substantially higher than for VKAs, total treatment costs could be reduced by increasing effectiveness and reducing the need for therapeutic monitoring [18]. Indeed, a Markov model was constructed using data from the EINSTEIN-DVT and EINSTEIN-PE trials suggesting that rivaroxaban was a cost-effective strategy compared with VKAs with an incremental cost effectiveness ratio (ICER) of GBP 7072 per QALY for PE [19]. Furthermore, in a cost-effectiveness analysis based on the HOKUSAI-VTE clinical trial data with a lifetime horizon, edoxaban was associated with 0.033 additional QALY and a low difference in costs of GBP 55 as compared to warfarin. Nonetheless, authors pointed out the necessity of using data coming from real-world studies rather than clinical trials because the balance between higher drug costs and the potential decrease of costs due to the reduction of VTE-related adverse events remains uncertain in current clinical practice [11,20].

The aim of this study was to assess the cost-effectiveness of oral anticoagulants, rivaroxaban versus VKAs, for the treatment of VTE in a real-life setting, the REMOTEV registry population.

## 2. Materials and Methods

### 2.1. Study Population, Setting and Location 

REMOTEV is an ongoing, prospective registry including all consecutive patients hospitalized in the Vascular Medicine Unit of Strasbourg University Hospital for acute deep vein thrombosis (DVT) and/or pulmonary embolism (PE) with a one year follow-up [21]. The present analysis focused on patients enrolled between 23 October 2013 (starting date of the registry) and 31 July 2015, aiming to evaluate the cost-effectiveness of two pharmacological strategies (DOACs, i.e., rivaroxaban, versus VKAs) after 6 months of follow-up. All patient data were gathered at the initial visit and via phone call at 6 months. Rivaroxaban was the only DOAC used in this study.

### 2.2. VTE Diagnosis

PE was diagnosed by either CT pulmonary angiogram (CTPA) or ventilation perfusion lung scan in case of contraindication to CTPA, while DVT was assessed by compression ultrasound (CUS) of lower limbs.

### 2.3. Comparators

We compared two strategies for the treatment of VTE patients: rivaroxaban and VKAs. Each patient was included in a treatment group according to the anticoagulant prescribed at hospital discharge.

### 2.4. Perspective

The analysis was conducted from the French healthcare provider perspective.

### 2.5. Time Horizon

The time horizon was 6 months after hospital discharge. 

### 2.6. Discount Rate

No discount rate was applied.

### 2.7. Outcomes’ Selection, Measurement and Valuation

We assessed the cost-effectiveness ratio of two strategies of oral anticoagulation for VTE treatment, i.e., rivaroxaban and VKAs, on the occurrence of the following outcomes: major and clinically relevant non-major (CRNM) bleeding, recurrent VTE and all-cause deaths at 6 months. Major and CRNM bleeding events were allocated according to the International Society on Thrombosis and Haemostasis (ISTH) criteria [22]. The evaluation criteria were prospectively assessed by senior physicians of the vascular medicine unit. ICER was the *primary endpoint* and was calculated as the difference in costs divided by the difference in efficacy between the two treatment strategies. The *secondary endpoints* were: (1) the hospital length of stay and (2) the number of hospital readmissions related to the index thromboembolic event.

### 2.8. Measurement and Valuation of Resources and Costs 

For the economic evaluation, the following data were retrospectively colligated: *(1) Costs items*: the number and unit costs of medications; the number and unit costs of blood tests; the number and costs of anticoagulant injections; the costs of the initial hospitalization and hospital readmissions during the 6-month follow-up period; and *(2) Effects items:* the number of adverse events (major and CRNM bleedings, recurrent VTE and all-cause deaths) during the 6-month follow-up.

### 2.9. Currency, Price Date, and Conversion 

The identification of the hospitalization and monitoring data were obtained from the local hospitals’ claims database. Hospitalizations were valued using the corresponding French disease related group (DRG) cost, adjusted by the length of stay and the number of days in ICU [16,23,24]. Ambulatory costs during the follow-up were valued using the statutory health insurance tariffs (Appendix A).

### 2.10. Statistical Analysis

Quantitative variables were presented as mean ± standard deviation, and categorical variables as number of cases (percentage). The chi-square or Fisher’s exact tests, ANOVA or Kruskall–Wallis were used as appropriate. There were no missing data concerning cost and effectiveness since we extracted data from the electronic health records and from the health insurance national database. We considered significant a test with a *p*-value lower than 0.05. All analyses were performed using R software version 4.0.3. (R Development Core Team 2020).

#### 2.10.1. Rationale and Description of Model 

To compare outcomes between treatment strategies in this non-randomized study, we used regression models weighted by inverse probability of the propensity score including demographic variables (age, sex), medical history (arterial hypertension, diabetes, stroke), laboratory tests (Cockcroft clearance of creatinine) and the sPESI score. For the missing data of these variables, a multiple imputation with chained equation was performed for each variable of the propensity score. We used a logistic regression model to model the effectiveness. For the costs, we used a log-linked Gamma generalized linear model.

#### 2.10.2. Analytics and Assumptions 

We calculated the ICER as the average increment of cost in the rivaroxaban compared to VKAs group divided by the average increment of effectiveness.

#### 2.10.3. Characterizing Uncertainty

The variability of the results was evaluated using a non-parametric bootstrap which provided multiple estimates of the ICER by randomly re-sampling the study population 1000 times. For each iteration, we computed propensity scores and new regression models. Results were presented as a scatter plot of 1000 ICERs on the cost-effectiveness plane.

#### 2.10.4. Characterizing Heterogeneity

We carried out a deterministic sensitivity analysis to investigate the independent effect of the following variables on the ICER: substituting costs of medications, costs of laboratory tests, and hospitalization costs with five times the base case for each parameter.

### 2.11. Ethics Approval, Data and Safety Monitoring

This study was conducted in accordance with Good Clinical Practice guidelines and the principles of the Declaration of Helsinki. The study was approved by the local Ethics Committee (CE 2019-14). A declaration of conformity was obtained from the Commission nationale de l’informatique et des libertés (CNIL) (agreement number 2208067v0). Patients were informed about the purpose of the registry and gave an oral consent to their participation according to the requirements of the local Ethics Committee. This study was registered on ClinicalTrials.gov (NCT03887806).

## 3. Results

### 3.1. Patient Characteristics at Baseline

From 23 October 2013 to 31 July 2015, 373 patients hospitalized for VTE were included in the REMOTEV registry and treated with either rivaroxaban or VKAs (Figure 1). 

The majority of patients (*n* = 279) were treated with rivaroxaban at hospital discharge, while VKAs were prescribed for 94 patients (63.1 ± 17.9 versus 71.3 ± 16.6 years old, *p* < 0.001). In the base-case population, stroke, arterial hypertension and chronic kidney disease were less frequent in the rivaroxaban group compared with the VKAs group. Furthermore, creatinine clearance was lower in VKAs-treated patients than in the rivaroxaban group (*p* = 0.012). Among PE patients, a sPESI score of 0 was more frequent in the rivaroxaban group (58% versus 39%, *p* < 0.05) (Table 1).

### 3.2. Study Evaluation Criteria 

#### 3.2.1. Effectiveness and Safety

In our population, fewer events occurred in rivaroxaban-treated patients as compared to those treated with VKAs, but differences were not statistically significant (Table 2). VTE recurrence and bleeding-related death were the leading causes of mortality with no difference between groups in the rivaroxaban- and the VKA-treated patients, respectively.

#### 3.2.2. Secondary Endpoints

The length of hospital stay was significantly shorter in the group treated by rivaroxaban than in the group treated by VKAs (6.47 ± 3.49 days versus 10.1 ± 6.80, *p* < 0.001) (Table 1).

The number of hospital readmissions in relation with the index VTE event was higher in the group treated by VKAs than rivaroxaban (19 (20.2%) versus 33 (11.2%), *p* = 0.042).

### 3.3. Propensity-Score Weighted Populations 

The inverse probability weighting from propensity scores allowed a better-standardized difference of means for the variables incorporated in the logistic model estimating the propensity score, i.e., after weighting, the absolute mean difference was inferior to 0.2 for all considered variables (Figure 2). 

### 3.4. Incremental Costs and Outcomes 

#### 3.4.1. Costs and Effectiveness

In our population, the average 6-month unadjusted costs were estimated at EUR 7721 (±6144) for VKAs and EUR 5201 (±3071) for rivaroxaban (Table 3).

The average effectiveness and the average costs in euros after inverse probability weighting are presented in Table 4. Rivaroxaban treatment was the dominant strategy, with a cost difference of EUR -2059 [95% CI −3582; −817] and a difference of efficacy of 0.005 [95% CI −0.06; 0.08] (Table 4).

#### 3.4.2. Effect of Uncertainty 

The set of ICERs estimated by the non-parametric bootstrap is presented as a scatterplot on the cost-effectiveness plane, with the VKAs treatment strategy as the reference. About 60% of these replications were located in the bottom right-hand quadrant, indicating a lower cost for greater effectiveness of rivaroxaban, translating into a 60% probability that rivaroxaban treatment strategy was dominant (Figure 3).

#### 3.4.3. Characterizing Heterogeneity

In the sensitivity analysis, rivaroxaban remained dominant for all deterministic sensitivity analyses with an estimated upper bound (five times the base case for each parameter) (Table 5).

## 4. Discussion

This non-randomized, prospective, registry-based study comparing rivaroxaban to VKAs standard therapy for the treatment of acute VTE showed that rivaroxaban was the dominant strategy in the cost-effectiveness analysis. Of note, in the unadjusted cohort, patients treated with VKAs were older, had more severe PE (higher sPESI scores) and more frequently had a chronic kidney disease. In order to overcome these differences, an inverse probability weighting from propensity scores was used and cost-effectiveness was assessed for each treatment strategy on weighted endpoints. Rivaroxaban proved efficient and cost-saving with 0.005 [95% CI −0.06; 0.08] events avoided and a cost reduction of EUR -2059 [95% CI −3582; −817] per patient compared to VKAs. The cost advantage for rivaroxaban was driven by a shorter duration of hospitalization and fewer hospital readmissions.

According to the most recent guidelines, DOACs are the first-line treatment for VTE [2,14]. It is now generally accepted that DOACs offer irrefutable pharmacological benefits compared to VKAs, such as a rapid onset of action, less drug-to-drug interactions and no laboratory monitoring, while the main limitation of VKAs was the unpredictable INRs requiring frequent blood monitoring From a clinical point of view, the main advantage of DOACs was the 40% (RR 0.60 (95% CI 0.41–0.88)) reduction in major bleeding, notably for intracranial localization and fatal bleeding, compared to VKAs that showed an estimated risk of major bleeding of 3% per year [25,26,27,28]. Furthermore, DOACs were non-inferior to VKAs for the prevention of VTE recurrence [26]. Indeed, the odds ratios for VTE recurrence range from 0.83 (95% CI 0.58–1.18) to 1.09 (95% CI 0.75–1.59) for DOACs as compared to warfarin [17].

However, DOACs have higher treatment costs than VKAs. Indeed, based on a National Institute for Health and Care Excellence (NICE) costing report, the 6-month treatment cost of DOACs was estimated at around GBP 400 as compared to GBP 210 for VKAs [17,29,30]. However, comprehension of the economic impact of VKAs and DOACs is more complex and the cost perspective is essential in the choice of VTE therapy. Nielsen at al. showed in a cost-analysis comparison from the Danish setting perspective that, when considering the narrowest spectrum of “drugs cost only”, VKAs were associated with lower costs than DOACs, but when including the economic impact of preventing recurrent VTE and limiting bleedings, apixaban and rivaroxaban resulted in marginally lower healthcare costs than VKAs [31]. Moreover, Spyropoulous et al. found that for morbidly obese VTE patients, average medical costs per patient per year were USD 2829 lower with rivaroxaban versus warfarin, due to the difference in hospitalization costs [32]. Nonetheless, in that study, no cost-effectiveness analysis was conducted and results cannot be generalized to all VTE patients. 

In contrast, Sterne et al., in an NHS cost-effectiveness analysis comparing DOACs to warfarin, revealed a poor difference of costs between the two strategies, with the lowest expected 6-month cost being for warfarin (GBP 19,651), followed by dabigatran, edoxaban, apixaban and then rivaroxaban, which was the most expensive treatment (GBP 19,753). The authors concluded that in cases where there was very low willingness to pay per QALY, warfarin was the most cost-effective treatment, but at GBP 20,000–30,000 per QALY thresholds, the probability that apixaban was the most cost-effective was approximately 0.54 [17]. Alternatively, for de Jong et al., there was an 85% probability for rivaroxaban being dominant compared with VKAs in the Netherlands, and 100% at a willingness-to-pay threshold of EUR 20,000/QALY [33]. In the same way, apixaban treatment was cost-effective in the UK compared to 6-month treatment with VKAs at an incremental cost-effectiveness ratio (ICER) of GBP 6692 per QALY based on a pharmacoeconomic analysis of the Amplify trial [34]. Sun at al., in a long-term cost-effectiveness analysis on the Chinese population including three DOACs (rivaroxaban, apixaban and dabigatran) and VKAs, found that rivaroxaban was the most cost-effective [35]. The systematic review of Al Mukdad et al. confirmed these results, revealing that apixaban was the most cost-effective strategy for VTE treatment followed by rivaroxaban, edoxaban and dabigatran [36].

### Strengths and Limitations

Our study is one of the few published medico-economic evaluations of rivaroxaban versus standard therapy with VKAs in VTE patients. Moreover, its real-life setting is an asset as highly selected clinical trial populations may not reflect actual VTE daily practice. However, the generalizability of our findings is constrained by the cost differences across countries. In the UK, the average drug costs were GBP 5.4 for a 6-month treatment with VKAs and ranged from GBP 378 to GBP 792 for a 6-month treatment with DOACs [37]. In the US, an 18-week treatment cost for DOACs was estimated at USD 2397 [38]. To account for these differences between healthcare systems, we performed a deterministic sensitivity analysis. When we varied costs to reflect health costs in different countries, DOACs remained cost-effective across broad settings. Although uncertainty was determined using a sensitivity analysis, our results should be interpreted within the setting of this analysis. The main limitation of this study was its observational nature, which may account for other factors than the treatment strategy, which could explain the observed differences between groups. In order to compensate for the considerable heterogeneity between groups in the unadjusted population, we performed an inverse probability weighting using a propensity score. Furthermore, the time horizon of this analysis was limited to 6 months, similar to the previously published clinical study [21]. Although the natural history of VTE could be longer than 6 months, the majority of costs concerned initial hospitalization. Moreover, this time horizon ensured a homogeneity of anticoagulant dosing regimens and is in line with the NICE supporting documentation on events occurring 90 days after hospital discharge [17].

Although DOACs represent a valuable alternative compared to warfarin for the treatment of VTE patients, according to cost-effectiveness analyses using published clinical trials, new data were necessary in order to perform a real-world practice cost-effectiveness analysis, instead of using clinical trial data to feed the model [20]. To our knowledge, this study was the first analysis confirming the cost-effectiveness of rivaroxaban compared to VKAs for the treatment of VTE using real-world data. These results provide significant new data for healthcare providers, confirming that DOACs are an efficient strategy for patients eligible for this treatment in a real-life setting. 

## 5. Conclusions

This study showed that in a real-life setting based on the REMOTEV registry analysis, DOACs are not only efficient and safe alternatives to VKAs, but also cost-saving, adding supplementary support to payers’ and physicians’ decision-making.

## Figures and Tables

**Figure 1 medicina-59-00181-f001:**
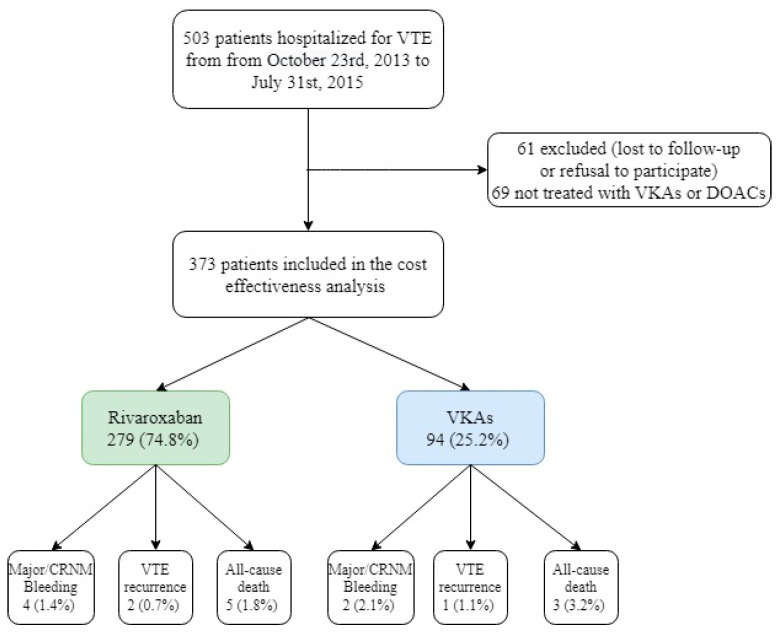
Study flowchart showing eligible patients’ selection. CRNM: clinically relevant non-major; DOACs: direct oral anticoagulants; VKAs: vitamin K antagonists; VTE: venous thromboembolism.

**Figure 2 medicina-59-00181-f002:**
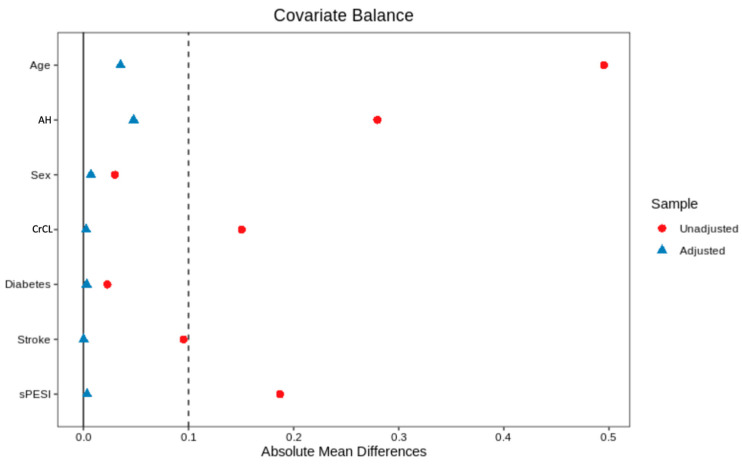
Standardized mean difference (SMD) before (red) and after (blue) inverse probability weighting from propensity scores. AH: arterial hypertension; CrCl: Cockcroft clearance of creatinine; sPESI: simplified Pulmonary Embolism Severity Index.

**Figure 3 medicina-59-00181-f003:**
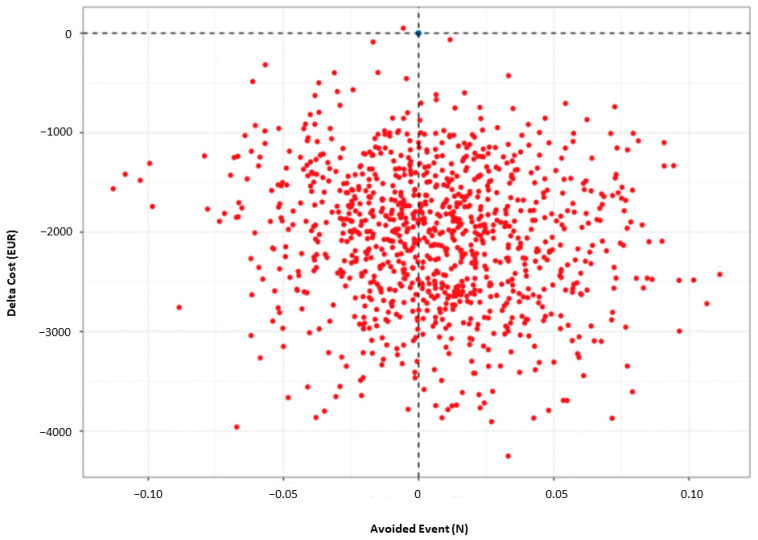
Scatter plot of incremental cost and effectiveness of rivaroxaban (red dots) compared to VKAs as the reference (blue dots). N: number.

**Table 1 medicina-59-00181-t001:** Patients’ characteristics at inclusion.

Characteristics		Rivaroxaban *n* = 279 M ± SD/*n*(%)	VKAs *n* = 94 M ± SD/*n*(%)	*p*
Age (years)		63.1 (±17.9)	71.3 (±16.6)	**<0.001**
Male sex		136 (49%)	43 (46%)	0.61
Medical history				
	Diabetes	42 (15%)	12 (13%)	0.59
	Arterial hypertension	61 (65%)	139 (50%)	**0.011**
	Stroke	15 (5.4)	14 (14.8)	**<0.001**
	Chronic kidney disease	11 (3.9%)	23 (24%)	**<0.001**
Admission laboratory tests				
	Ddimers (ug/L)	5100 (±5295)	5459 (±5045)	0.59
	CrCl (mL/min)	101 (±46.8)	83.1 (±62.2)	**0.012**
Diagnosis of pulmonary embolism		251 (90)	87 (92.6)	0.46
sPESI = 0		146 (58%)	34 (39%)	**0.002**
Length of hospital stay (days)		6.47 (±3.49)	10.1 (±6.80)	**<0.001**

CrCl: Cockcroft clearance of creatinine; M: mean; *n*: number; SD: standard deviation; sPESI: simplified pulmonary embolism severity index; VKAs: vitamin K antagonists; VTE: venous thromboembolism; bold indicates a significant *p*-value.

**Table 2 medicina-59-00181-t002:** Six-month outcomes.

Events	Rivaroxaban *n* = 279 *n* (%)	VKAs *n* = 94 *n* (%)	*p*
Major or clinically relevant non-major bleeding	4 (1.4)	2 (2.1)	0.23
Recurrent VTE	2 (0.7)	1 (1.1)	0.3
All-cause deaths VTE-related death Bleeding-related death	5 (1.8) 4 (1.4) 0	3 (3.2) 0 2 (2.1)	0.42

*n*: number; VKAs: vitamin K antagonists; VTE: venous thromboembolism.

**Table 3 medicina-59-00181-t003:** Rivaroxaban versus VKAs average costs per patient for the unadjusted population.

Costs (Euros)	Rivaroxaban *n* = 279 M ± SD	VKAs *n* = 94 M ± SD
Drugs	EUR 350 (±0)	EUR 178 (±175)
Laboratory tests	EUR 10.5 (±0)	EUR 331 (±30.8)
Hospitalization	EUR 4397 (±2478)	EUR 5715 (±3497)
Hospital Readmission	EUR 444 (±1669)	EUR 1497 (±4974)
Total costs	EUR 5201 ± 3071)	EUR 7721 (±6144)

DOACs: direct oral anticoagulants; M: mean; SD: standard deviation; VKAs: vitamin K antagonist.

**Table 4 medicina-59-00181-t004:** Costs and effectiveness for each anticoagulant treatment (weighted endpoints).

	Rivaroxaban *n* = 279	VKAs *n* = 94	Difference
Cost per patient (95% CI)	EUR 5662 [6606; 9060]	EUR 7721 [5130; 6304]	EUR 2059 [−3582; −817]
Effectiveness (95% CI)	0.0586 [0.0114; 0.126]	0.0638 [0.0208; 0.109]	0.00527 [−0.0606; 0.0761]

CI: confidence interval; *n*: number; VKAs: vitamin K antagonists.

**Table 5 medicina-59-00181-t005:** Deterministic sensitivity analysis.

Determinant of Analysis	Cost		Effectiveness		Difference of Cost	Difference of Effectiveness
	Riva	VKAs	Riva	VKAs	Riva vs. VKAs	Riva vs. VKAs
Base case	EUR 5662	EUR 7721	0.0586	0.0638	EUR 2059	0.00527
Drugs	EUR 7061	EUR 8434	0.0586	0.0638	EUR 1373	
Laboratory tests	EUR 5704	EUR 9046	0.0586	0.0638	EUR 3342	
Hospitalization	EUR 26870	EUR 36570	0.0586	0.0638	EUR 9700	

ICER: incremental cost effectiveness ratio; Riva: rivaroxaban; VKAs: vitamin K antagonists; vs.: versus.

## Data Availability

Not applicable.

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
