# Peer review of "A Real-World Cost-Effectiveness Analysis of Rivaroxaban versus Vitamin K Antagonists for the Treatment of Symptomatic Venous Thromboembolism: Lessons from the REMOTEV Registry"

_medicina, 2023, doi:10.3390/medicina59010181_

Round 1

Reviewer 1 Report

Manuscript ID: medicina-2127126

A real-world cost-effectiveness analysis of rivaroxaban versus vitamin K antagonists for the treatment of symptomatic venous thromboembolism : lessons from the REMOTEV registry

Sabrina Kepka , Elena-Mihaela Cordeanu * , Kevin Zarca , Anne-Sophie Frantz , Patrick Ohlmann , Emmanuel Andres , Pascal Bilbault , Isabelle Durand-Zaleski , Dominique Stephan

Very interesting and well written paper on an important topic. However, I have some comments and suggestions which may improve the quality of this manuscript.

The present study evaluates the cost-effectiveness of rivaroxaban compared with vitamin K antagonists in patients with venous thromboembolism.

It would be appropriate to replace the word DOACs with rivaroxaban throughout the manuscript, including figures and tables, given that it is the only direct anticoagulant that is taken into consideration.

It would be advisable to check the appropriateness of the sample size. 279 patients (74.8%) were enrolled in the rivaroxaban group, while only 94 (25.2%) in the VKAs group. This large difference in the two populations studied could significantly affect the results in terms of efficacy and safety.

The various studies cited by Kepka and colleagues do not take into consideration the heterogeneity of the enrolled populations. Especially the different number of patients with active cancer.

Author Response

Answers to Reviewers

Strasbourg, the 10th of January 2023

Dear Reviewers,

Dear Editors, 

We are pleased to submit to the Medicina Journal a revised version of the manuscript “A real-world cost-effectiveness analysis of rivaroxaban versus vitamin K antagonists for the treatment of symptomatic venous thromboembolism : lessons from the REMOTEV registry” (Manuscript ID: medicina-2127126) by Kepka and coworkers.

We believe that Reviewers suggestions have helped us improve the quality of the paper and hope that you will find it suitable for publication.

  1. General comments:

Our answers to Reviewers’ comments appear in blue (in this document) whereas all changes in the manuscript appear in red.

We thank Reviewers for the constructive comments and we are grateful for their remarks that allowed us to improve our manuscript readability despite certain limits related to its observational nature and the limited number of patients included.

In order to comply to Editors’ requirements based on the similarity report, several wording changes were performed and appear in red in the text. All these changes don’t affect the meaning nor interfere with the understanding of the text but, we hope, will reduce text similarities with previous publications of the authors or articles on the same subject. Of note, among the similarities, appear for instance, the scientific belonging of the authors or the footer of the article (see after) but also classical terms used together such as “Oral anticoagulants (direct oral anticoagulants (DOACs) or vitamin K antagonists (VKAs))” that could not be changed.

We are now going to address all reviewers comments:

  1. Reviewer(s)' Comments to Authors:

Reviewer 1

We thank R1 for his/her comments and overall good appreciation.

Reviewer 1 comment 1. It would be appropriate to replace the word DOACs with rivaroxaban throughout the manuscript, including figures and tables, given that it is the only direct anticoagulant that is taken into consideration.

R1A1: We agree with this text modification and made the necessary changes.

Reviewer 1 comment 2. It would be advisable to check the appropriateness of the sample size. 279 patients (74.8%) were enrolled in the rivaroxaban group, while only 94 (25.2%) in the VKAs group. This large difference in the two populations studied could significantly affect the results in terms of efficacy and safety.

R1A2: We fully agree with R1 remarks concerning the proportion difference between the two populations. It is however impossible to obtain in an observational cohort study similar populations for DOACs and VKAs as DOACs are prescribed in more than 80% of VTE patients and VKAs are reserverd nowadays to DOACs contraindications.

However this analysis was conducted on the initial population of the REMOTEV registry (2013-2015) when VKAs were still prescribed to a larger amount of VTE patients and not only to those that could not receive a DOAC allowing comparison between groups. REMOTEV is nonetheless an observational study and in order to compare non-randomized groups of patients, treatment selection bias needs to be dealt with. In the first publication of REMOTEV registry, the same population was adjusted according to a propensity score matching on several covariates that were deemed influential in terms of anticoagulant choice: age, eGFR, weight, quadrimester of inclusion in the registry but this matching technique conducted to discarding some unmatched patients from the VKA group. The propensity score used in the present analysis is more appropriate for a medico economical analysis and avoids the loss of patients. The inverse probability of treatment weighting (IPTW) allowed adjustement for age, sex, medical history (hypertension, diabetes, stroke, renal function) and PE severity (sPESI score) with a standardized difference between groups <0.20 after adjustment.

Reviewer 1 comment 3: The various studies cited by Kepka and colleagues do not take into consideration the heterogeneity of the enrolled populations. Especially the different number of patients with active cancer.

R1A3: In REMOTEV, active cancer patients were treated according to guidelines by LMWH for the period concerned by this data extraction (2013-2015). So, all active cancers known at the time of patient’s hospital discharge (previously known cancers and newly discovered ones) were de facto excluded from our study population as these patients were not on oral anticoagulation. Four patients initially treated with DOACs or VKAs at discharge had a diagnosis of occult cancer during FU and were included in the present analysis. However, these 4 patients did not present any of the main outcomes during the 6 month FU. Until recently all international guidelines favoured LMWH during the first six months of anticoagulant therapy following cancer-associated VTE. Therefore, we believe that a pertinent comparison of anticoagulant cost-effectiveness for cancer patients would concern DOACs versus LMWH as VKAs are not indicated in cancer-associated thrombosis.

In hope that our revised manuscript will comply to reviewers’ requests,

With my best regards.

Sincerely yours.

Elena-Mihaela CORDEANU (for the Authors)

Reviewer 2 Report

This prospective study compared both cost and effectiveness between DOACs (rivaroxaban) and VKA (Vit K antagonist) in treating venous thromboembolism (n=373) and authors concluded that DOACs are not only efficient and safe VKAs alternatives for eligible VTE patients, but also cost-saving. Overall this study is interesting and have profound clinical importance. I have the following concerns.

1. In the Introduction, authors may add more background on the contributors to VTE occurrence. i.e schizophrenia patients are reported to more easily display VTE (PMID: 31688052). People with COVID-19 are also reported to display VTE (PMID: 32374815). These information would increase the readability of this study.

2. In the "2.6. Selection, measurement and valuation of outcomes", authors did not state how the 373 patients were diagnosed with VTE and the how many patients died in the followed 6-months (as well as their detailed cause of death). Particularly, is there any difference between the mortality of the two treatments?

3. In Table 1, it is clear that patients receiving DOACs treatment is significantly younger than those receiving VKA. This would undoubtedly lead to case selection bias. Authors should comment on this. Also, I would suggest authors list out more medical history in Table 1, such as whether these patients had any mental illness or COVID-19.

4. Discussion is appropriate. The strengths and limitations could be shortened.

Author Response

Answers to Reviewers

Strasbourg, the 10th of January 2023

Dear Reviewers,

Dear Editors, 

We are pleased to submit to the Medicina Journal a revised version of the manuscript “A real-world cost-effectiveness analysis of rivaroxaban versus vitamin K antagonists for the treatment of symptomatic venous thromboembolism : lessons from the REMOTEV registry” (Manuscript ID: medicina-2127126) by Kepka and coworkers.

We believe that Reviewers suggestions have helped us improve the quality of the paper and hope that you will find it suitable for publication.

  1. General comments:

Our answers to Reviewers’ comments appear in blue (in this document) whereas all changes in the manuscript appear in red.

We thank Reviewers for the constructive comments and we are grateful for their remarks that allowed us to improve our manuscript readability despite certain limits related to its observational nature and the limited number of patients included.

In order to comply to Editors’ requirements based on the similarity report, several wording changes were performed and appear in red in the text. All these changes don’t affect the meaning nor interfere with the understanding of the text but, we hope, will reduce text similarities with previous publications of the authors or articles on the same subject. Of note, among the similarities, appear for instance, the scientific belonging of the authors or the footer of the article (see after) but also classical terms used together such as “Oral anticoagulants (direct oral anticoagulants (DOACs) or vitamin K antagonists (VKAs))” that could not be changed.

We are now going to address all reviewers comments:

  1. Reviewer(s)' Comments to Authors:

Reviewer 2.  

We thank R2 for his/her comments and overall good appreciation.

Reviewer 2 comment 1. In the Introduction, authors may add more background on the contributors to VTE occurrence. i.e schizophrenia patients are reported to more easily display VTE (PMID: 31688052). People with COVID-19 are also reported to display VTE (PMID: 32374815). These information would increase the readability of this study.

R2A1:  We have added the references to the text. Psychiatric conditions have indeed been associated with a higher VTE risk, but not higher risk of complications during anticoagulation treatment. As our study concerns the period from 2013 to 2015, COVID-19 was not repertoriated among risk factors.

Respecting R2 suggestion we have added in the background chapter of our article the following text and integrated the two references (Lines 50-52):

Certain risk factors have been associated with higher risk of VTE occurrence: immobilization, surgery, active can-cer, estrogen-based therapies, certain psychiatric conditions, acute and chronic inflammatory states and more re-cently COVID-19 infection [2–4].

Reviewer 2 comment 2. In the "2.6. Selection, measurement and valuation of outcomes", authors did not state how the 373 patients were diagnosed with VTE and the how many patients died in the followed 6-months (as well as their detailed cause of death). Particularly, is there any difference between the mortality of the two treatments?

R2A2: Indeed, several clinical data were not detailed as this was medico economical analysis of a population for which the clinical study has already been published. In order to facilitate the understanding of the methodology for readers that are not aquainted with our clinical study and to respond to R2 remark, we have added the following statement in the Methods chapter:

(Lines 84-86) VTE diagnosis

PE was diagnosed by either CT pulmonary angiogram (CTPA) or ventilation perfusion lung scan in case of contraindication to CTPA while DVT was assessed by compression ultrasound (CUS) of lower limbs.

In respect to the mortality, all-cause deaths appear in Table 2. We have added two lines to Table 2 and a comment on the causes of death to the text that accompanies Table 2:

Events

Rivaroxaban

N=279

N(%)

VKAs

N=94

N(%)

p

Major or clinically relevant non-major bleeding

4 (1.4)

2 (2.1)

0.23

Recurrent VTE

2 (0.7)

1 (1.1)

0.3

All-cause deaths

    VTE-related death

    Bleeding-related death

5 (1.8)

4 (1.4)

0

3 (3.2)

0

2 (2.1)

0.42

N: number; VKAs: vitamin K antagonists; VTE venous thromboembolism

(Lines 180-181) VTE recurrence and bleeding-related mortality were the leading causes of death with no difference between groups in the rivaroxaban and the VKA treated patients, respectively.

Reviewer 2 comment 3. In Table 1, it is clear that patients receiving DOACs treatment is significantly younger than those receiving VKA. This would undoubtedly lead to case selection bias. Authors should comment on this. Also, I would suggest authors list out more medical history in Table 1, such as whether these patients had any mental illness or COVID-19.

R2A3: Age as well as other comorbidities were more frequent in the VKA-treated patients. This is why, age was one of the covariates included in the inverse probability weighting. Medical history was detailed in the clinical paper of the same study population and as this was a medico-economical study we selected we deemed pertinent for this analysis and tried to avoid redudence with our previous publication (reference no 25 of the article). As for psychiatric pathologies (the registry did not distinguish among pathologies), they were 20.8% (=58) in the rivaroxaban group and 26.6% (n=25) in the VKA group, p=0.24. Moreover, our population was not concerned by the COVID-19 epidemic as the study period ended in 2015.

Reviewer 2 comment 4. Discussion is appropriate. The strengths and limitations could be shortened.

R2A4: We have reduced the strengths and limitation section as suggested by R2 from 437 to 365 words. All changes appear in red in the text.

In hope that our revised manuscript will comply to reviewers’ requests,

With my best regards.

Sincerely yours.

Elena-Mihaela CORDEANU (for the Authors)
